# Investigating Unfavorable Factors That Impede MALDI-TOF-Based AI in Predicting Antibiotic Resistance

**DOI:** 10.3390/diagnostics12020413

**Published:** 2022-02-05

**Authors:** Hsin-Yao Wang, Yu-Hsin Liu, Yi-Ju Tseng, Chia-Ru Chung, Ting-Wei Lin, Jia-Ruei Yu, Yhu-Chering Huang, Jang-Jih Lu

**Affiliations:** 1Department of Laboratory Medicine, Chang Gung Memorial Hospital at Linkou, Taoyuan City 333423, Taiwan; hsinyaowang@cgmh.org.tw (H.-Y.W.); weitinglin66@cgmh.org.tw (T.-W.L.); st33351995@cgmh.org.tw (J.-R.Y.); 2Ph.D. Program in Biomedical Engineering, Chang Gung University, Taoyuan City 333323, Taiwan; 3Department of Anesthesiology, Chang Gung Memorial Hospital at Linkou, Taoyuan City 333423, Taiwan; mpq192@cgmh.org.tw; 4Department of Computer Science, National Yang Ming Chiao Tung University, Hsinchu 300093, Taiwan; yijutseng@cs.nycu.edu.tw; 5Department of Computer Science and Information Engineering, National Central University, Taoyuan City 320317, Taiwan; jjrchris@g.ncu.edu.tw; 6Division of Pediatric Infectious Diseases, Department of Pediatrics, Chang Gung Memorial Hospital, Taoyuan City 333423, Taiwan

**Keywords:** methicillin-resistant *Staphylococcus aureus*, matrix-assisted laser desorption/ionization time-of-flight, antibiotic susceptibility test, artificial intelligence

## Abstract

The combination of Matrix-Assisted Laser Desorption/Ionization Time-of-Flight (MALDI-TOF) spectra data and artificial intelligence (AI) has been introduced for rapid prediction on antibiotic susceptibility testing (AST) of *Staphylococcus aureus*. Based on the AI predictive probability, cases with probabilities between the low and high cut-offs are defined as being in the “grey zone”. We aimed to investigate the underlying reasons of unconfident (grey zone) or wrong predictive AST. In total, 479 *S. aureus* isolates were collected and analyzed by MALDI-TOF, and AST prediction and standard AST were obtained in a tertiary medical center. The predictions were categorized as correct-prediction group, wrong-prediction group, and grey-zone group. We analyzed the association between the predictive results and the demographic data, spectral data, and strain types. For methicillin-resistant *S. aureus* (MRSA), a larger cefoxitin zone size was found in the wrong-prediction group. Multilocus sequence typing of the MRSA isolates in the grey-zone group revealed that uncommon strain types comprised 80%. Of the methicillin-susceptible *S. aureus* (MSSA) isolates in the grey-zone group, the majority (60%) comprised over 10 different strain types. In predicting AST based on MALDI-TOF AI, uncommon strains and high diversity contribute to suboptimal predictive performance.

## 1. Introduction

Artificial intelligence (AI) has been successfully applied in a variety of medical practices, with faster diagnostic speed and similar accuracy compared to expert judgements [1]. However, the confidence of prediction never reaches 100% [2]. The predictive uncertainty might have multiple sources, such as missing information, bias, noise, and dataset shift [3]. In medical AI, especially for life-critical decision making, reporting the uncertainty of prediction is required [3,4,5]. A key to medical AI success is to calibrate human trust by providing a confidence score in the model on a case-by-case basis [5,6]. By providing the uncertainties to decision makers, the abilities of machines and humans are combined and the prediction performance can be enhanced [2,3,5]. Understanding the uncertainties of AI is crucial for implementation in a clinical setting. However, the related issues have not yet been widely investigated.

Methicillin-resistant *Staphylococcus aureus* (MRSA) is causing a major public health problem with resistance to commonly used antibiotics, varying epidemiology of infection, and increased morbidity and mortality [7,8,9]. Rapid and correct administration of antibiotics, such as vancomycin, teicoplanin, or linezolid, is the key to successful treatment [10]. Antibiotic susceptibility testing (AST) is the gold standard guiding the administration of these anti-infective agents [7,11]. However, this culture-based method can cause considerable delay in prescribing effective antimicrobial treatment, because it takes an additional 3 to 4 days after specimen collection to produce the susceptibility reporting results [7,12]. A rapid assessment of antibiotic resistance can optimize antimicrobial treatment, reducing unnecessary antibiotic use, and avoiding development of antibiotic resistance. A novel method to accelerate antimicrobial susceptibility testing was developed and validated in our previous studies [13,14], with the combination of large-scale Matrix-Assisted Laser Desorption/Ionization Time-of-Flight (MALDI-TOF) mass spectra data and AI. This approach harnesses the powerful pattern recognition ability of AI in exploring the patterns of MS spectra. MS spectra comprise several hundreds of peaks, which mostly represent peptides or proteins. As a phenotype, antibiotic resistance is caused by specific proteins. Thus, the peptides or proteins that deliver antibiotic resistance can be detected by MALDI-TOF MS and the specific patterns of proteins and peptides can be detected by AI algorithms. In the studies, Wang et al. collected around 5000 mass spectrometry (MS) spectra of unique *S. aureus* isolates and identified 200 peaks on the MS spectra, which present remarkable differences between MRSA and methicillin-susceptible *S. aureus* (MSSA). These peaks serve as the marker features for the construction of the AST predictive model. Random forest was used as the machine learning classification algorithm for its outstanding prediction performance in an independent test, with the area under the receiver operating characteristic curve (AUC) at 0.845. Based on the AI model and MS spectrum, only a few minutes are needed to obtain the preliminary AST of oxacillin. The studies demonstrated that incorporating an AI method into a large-scale dataset of clinical MS spectra would recognize antibiotic-resistant bacteria strains in a much shorter time and lead to a more favorable clinical outcome.

Correct prediction leads to immediate and appropriate antimicrobial treatment, while an incorrect preliminary result may misguide and delay the administration of antibiotics. In order to accurately guide clinical decisions, lowering the wrong prediction rate is necessary. Preliminary AST was determined by the prediction probability calculated by the AI model. In the design of the AI model, the predictive probability ranges from 0 to 1. The cut-off was set to 0.48, whereas an isolate with probability lower than 0.48 is predicted as MSSA, and MRSA is predicted when probability is higher than 0.48. In the deployment of the AI model [15], we found that wrong predictions frequently occurred in the probability range between 0.40 and 0.48. (See Appendix A.) The predictions with probability in this range were not released to a clinical setting. The range was defined as the grey zone (see Appendix A) and a probability of 0.40 was set as the low cut-off, whereas 0.48 was the high cut-off.

Grey zone is a very common technique that is used to improve test accuracy in many clinical laboratory tests [16,17,18]. However, the grey zone diminishes the benefit brought by the MALDI-TOF-based AI model because predictive AST is not provided for the cases in the grey-zone group. Thus, in this study we aimed to investigate the possible factors that are associated with grey zone predictions and wrong predictions by the AI model. The novelty of this study is to analyze the factors attributed to the uncertainties of MALDI-TOF-based AI. This investigation helped us establish the disadvantages, or blind spots, of the MALDI-TOF-based AI model. Based on the results, rapid AST with MALDI-TOF-based AI can be understood by clinical physicians. Thus, rapid AST-guided management would be accepted and implemented more in clinical settings.

## 2. Materials and Methods

### 2.1. Scheme of the Study

The study aimed to identify and further analyze *S. aureus* isolates of a grey-zone group and a wrong-prediction group. A schematic illustration of this study can be seen in Figure 1, comprising three steps: (1) sample collections and MALDI-TOF spectra measurement; (2) AST prediction with AI model and final AST report; and (3) analysis of grey-zone and wrong-prediction samples. First, samples were collected in the clinical microbiology laboratory of Linkou Chang Gung Memorial Hospital. The cultured bacterial samples were analyzed by MALDI-TOF MS for identification of bacterial species. Preliminary AST was predicted by the AI model after inputting preprocessed MS spectra. Predictive probability ranging between 0.40 and 0.48 was defined as the “grey zone” [16,17,18] and was not to be used in a clinical setting. Specimen types, MALDI-TOF MS spectra, and phenotypic susceptibility test reports were collected. MLST was also identified for further investigation for unconfident AST predictions (i.e., grey zone). The cases with wrong AST prediction were also analyzed via the same method as the grey-zone group.

### 2.2. Sample and Mass Spectra Collection

The study was approved by the Institutional Review Board of the Chang Gung Medical Foundation (No. 202000694B1, Date of Approval: 18 April 2020). Clinical specimens were collected at the Linkou branch of Chang Gung Memorial Hospital (CGMH) from August to October 2020 and sent to CGMH clinical microbiology laboratory. The specimen types included wound, respiratory tract (i.e., sputum, nasopharyngeal swab, and bronchoalveolar lavage), blood, tissue, urinary tract, sterile body fluid (i.e., ascites, pleural effusion, synovial fluid, dialysates, and cerebrospinal fluid), and others. Cultures were obtained by routine method in CGMH clinical microbiology laboratory [13,14]. Single colonies on agar plates were chosen for bacterial species identification. *S. aureus* was identified according to colony morphology, coagulase test, and MALDI-TOF MS (Bruker Daltonics GmbH, Bremen, Germany) [19]. Once the MS spectra were generated and were identified as *Staphylococcus aureus*, they underwent preprocessing and feature extraction [14] as preparation for inputting to the AST prediction model.

### 2.3. Preliminary AST with AI Model and Traditional AST

We applied the AST predictive models that we developed and validated in previous studies [13,14]. After applying the preprocessed *S. aureus* MS spectra to the AI model, preliminary AST was predicted within one minute. The prediction results were presented with peak number, probability, and preliminary AST. The peaks in mass spectra can be representative of ribosomal proteins specific to species, and can serve as biomarkers for species identification [20,21]. Peak number represents the quality of input MS spectra. Predictive probability served as the basis of classification, as previously mentioned. For those with a probability range from 0.40 to 0.48, grey zone was assigned but would not be reported for clinical usage. If the probability was >0.48, the sample would be classified as MRSA; if it was <0.4, it would be predicted as MSSA. Traditional ASTs such as the cefoxitin paper disc method and broth microdilution method were performed to determine the susceptibility of *S. aureus* to oxacillin. The broth microdilution method was basically performed on specimens from blood and sterile body fluid, while the cefoxitin paper disc method was conducted on other types of specimens. The interpretation of AST was based on Clinical and Laboratory Standards Institute (CLSI) guidelines. Both methods are the standard CLSI-endorsed methods for determining the susceptibility of *S. aureus* to oxacillin.

### 2.4. Analysis of Grey-Zone and Wrong-Prediction Cases

For further understanding cases in the grey-zone group and wrong-prediction group, demographic information, MALDI-TOF MS spectra, predictive results, and traditional AST reports were reviewed. Regarding traditional ASTs, we recorded minimal inhibitory concentration of oxacillin or diffuse zone diameter of cefoxitin paper disc. For molecular characterization, we used multilocus sequence typing (MLST) for strain typing of the *S. aureus* isolates [22]. Sequence type was assigned based on the sequence allelic profiles at the seven loci, via the MLST database [23].

### 2.5. Statistical Analysis

Continuous variables were expressed as the means and standard deviations, categorical variables were documented as numbers and percentages, and nonparametric dependent variables were noted as medians and interquartile range. Student’s t test was used for continuous variables, and chi-square was used for categorical variables. ANOVA was used for mean comparison between more than two groups’ means, and a post hoc test (Scheffe) was performed if there was a statistically significant result. The Kruskal–Wallis test was used for nonparametric dependent variables with more than two groups, and Dunn’s multiple comparison test was used for a significant Kruskal–Wallis test. *p*-values were calculated and documented as two-sided, and a null hypothesis was rejected if the *p* value was smaller than or equal to 0.05. Specifically, we were testing whether specimen types were factors that would affect the predictive results. The *p*-value was used in the statistics to estimate whether the composition of specimen types was different between the predictive groups. A *p*-value of 0.05 was used as the cut-off in the study. When the *p*-value was less than 0.05, then a significantly different composition was detected. Subsequently, a post hoc analysis was conducted on the specific specimen type that was distributed differently across predictive groups. All analyses were performed with SPSS Statistics for Windows, version 28.0 (Statistical Product and Service Solutions, IBM Corp., Armonk, NY, USA).

## 3. Results

### 3.1. Investigating the Association between the Demographic Information and the AI Predictive Results

In the study, we aimed to investigate factors associated with unconfident prediction (i.e., grey zone) or wrong prediction. First, we tested the association between demographic information and predictive results. Table 1 shows the specimen types of all 479 cases. Of the 479 collected samples, 401 were in the correct-prediction group, 56 were in the grey-zone group, and 22 were in the wrong-prediction group. Pus was the major specimen type in the three groups (46.4–68.1%). The correct-prediction group had significantly more respiratory tract specimens, while the grey-zone group contained significantly more sterile body fluid specimens than the others (*p* = 0.006). For MSSA isolates, the correct-prediction group had significantly more respiratory tract specimens, and the grey-zone group had significantly more blood and sterile body fluid samples (*p* = 0.008). On the other hand, for MRSA, there was no specimen type difference between the groups.

### 3.2. Investigating the Association between the Quality of Mass Spectrum and AI Predictive Results

Second, the initial quality of the mass spectra in different prediction groups was examined. Figure 2 shows the peak numbers of the MALDI-TOF mass spectra in different groups. The peak numbers of spectra in different groups were consistently around 120. No difference in peak numbers was detected between correct-prediction group, grey-zone group, and wrong-prediction group (see Appendix A). The results indicated that the quality of MALDI-TOF mass spectra was comparable between the different groups.

### 3.3. Investigating the Association between the ASTs and AI Predictive Results

Third, we examined the AST results of oxacillin in different groups. Table 2 presents the mean zone size of cefoxitin disc diffusion test for *S. aureus* isolates. The mean zone size for MRSA in the correct-prediction group was 10.78 ± 3.49 mm; for the grey-zone group it was 11.4 ± 3.95 mm; for the wrong-prediction group, it was 14.33 ± 3.01 mm. ANOVA was performed and showed a significant difference in zone size (*p*-value = 0.004). Post hoc test (Scheffe) showed that the MRSA isolates in the wrong-prediction group had a significantly larger zone diameter than the MRSA isolates in the correct-prediction group. For MSSA, no significant difference was noted between the three groups. Table 3 shows the MIC of oxacillin using the broth microdilution method. There was no significant MIC difference between groups.

### 3.4. Investigating the Association between Strain Types and AI Predictive Results

Fourth, we analyzed the strain types (MLST) for the *S. aureus* isolates of the grey-zone group (Table 4). The isolate number of MRSA (*n* = 10) was much lower than that of MSSA (*n* = 46). For MSSA isolates in the grey-zone group (*n* = 46), many more strain types were identified. In total, 15 different types were identified. MSSA ST15 accounted for the highest percentage (41%) in the group. The remaining 59% MSSA in the grey-zone group comprised 14 different types. In the group, two MSSA isolates could not be typed. For MRSA isolates in the grey-zone group (*n* = 10), six types were identified. MRSA ST1232 accounted for the highest percentage (40%) in the group, followed by ST59 (20%). Only one isolate was identified for ST6954, ST239, ST30, and ST1.

The compositions of MRSA and MSSA in the grey-zone group were also compared with the molecular epidemiology published in the previous studies [24,25,26,27,28,29]. The top five types of MSSA aside from ST1 (i.e., ST15, ST188, ST7, and ST97) in the grey-zone group were also reported in the previous studies (Figure 3) [24,25]. By contrast, for the MRSA isolates in the grey-zone group, ST1232 and ST6954 were not reported as the major circulating strain type in the previous studies (Figure 4) [27,28,29].

## 4. Discussion

Specimen type may affect the predictive performance of AI models. Certain types of specimens tend to be infected by specific strain types of microorganism. Consequently, over-concentration of specific specimens would be associated with only a few specific strain types. A low number of classifications would simplify the classification problem for AI models, and, theoretically, that will elevate the predictive performance. We examined the specimen types for the three groups (i.e., correct-prediction group, grey-zone group, and wrong-prediction group). The results revealed that MSSA isolated from respiratory tract specimens tended to have correct AST prediction for oxacillin (Table 1). By contrast, MSSA isolated from blood or sterile body fluid was associated with a higher chance of grey-zone prediction (Table 1). MSSA isolates in the grey-zone group had more diverse sequence types than MRSA (Table 4). Other studies also revealed more heterogeneous MSSA lineages and wide genotypic diversity [24,25]. When the diversity of MSSA lineage is the reason for the uncertainty of the AST prediction, we hypothesize that MSSA in the respiratory tract was more homogeneous than MSSA from another specimen type. In this study, we only investigated the factors associated with grey-zone prediction and wrong prediction. The heterogeneity of *S. aureus* in different specimen types has not been well established. For correct predictions of respiratory tract specimens, it would be useful to investigate the underlying reasons.

We also examined the association between quality of mass spectra and the predictive results. The peaks in mass spectra can be representative of ribosomal proteins that were specific to species, and can serve as biomarkers for species identification [20,21]. Rich peak content provides adequate information for highly efficient species identification [30]. Peak numbers were analyzed to establish whether the AST prediction was affected by the peak content of MS spectra. The peak detection of MS spectra would be affected by many steps during MS spectra preparation, including sample collection, cultivation, subculture, incubation, colony selection, plate smearing, and even the condition of Microflex LT mass spectrometer [14]. There was no significant difference in peak numbers between the grey-zone, wrong-prediction, and correct-prediction groups (Figure 2). The peak content of the mass spectra was comparable between groups. Thus, we discovered that quality of mass spectrum was not a factor that was associated with the uncertain or wrong prediction of AST.

The nature of the drug resistance level is another factor that leads to an uncertain or wrong prediction in AST. For isolates with very high or very low antibiotic MIC, the AI model would perform well. By contrast, when the antibiotic MIC is close to the cut-off that discriminates between resistant and susceptible results, the cases would be difficult for the AI model to predict well. The zone size of the MRSA in the wrong-prediction group had a significantly larger zone diameter than the MRSA the correct-prediction group (Table 2). Resistance to a specific antimicrobial may require a complex mechanism rather than depending on the expression of a single gene. Different mechanisms may contribute to phenotypic diversity. One study discovered inconsistencies in antibiotic-resistance phenotypes and genotypes [31]. Some strains carry a drug resistance gene that is susceptible to the corresponding antibiotic, while some have drug resistance genes that are not expressed. This phenomenon may affect our predictive model’s ability to provide correct preliminary AST. In this study, the MRSA in the wrong-prediction group with a larger zone diameter may carry drug-resistance genes that were suboptimally expressed. Low-expression proteins or peptides may be undetectable with the standard protocol of MALDI-TOF mass. Thus, the mass spectra of these MRSA cases may lead the ML model to provide the wrong preliminary AST.

The composition of strain types could be the predominant factor for grey-zone predictions based on our results. In grey zone, the majority was MSSA (82.1%) (Table 4), indicating that a confident prediction for MSSA is more difficult. As previously mentioned, studies have shown that MSSA has more complicated genotypic diversity [24,25,26]. According to epidemiological investigations in Taiwan [24,25], shown in Figure 3, the predominant clones of MSSA infection were ST188, ST15, ST7, and ST97, but the remaining one-third of MSSA samples comprised other types. The molecular characteristics of MSSA in the grey zone showed comparable results where ST15 and ST188 were the major strain types, the other third comprised many different types (Figure 3). This heterogeneity of the MSSA lineage would also exist in the training dataset for the machine learning model. The AI model would have suboptimal learning for the diverse strain types because only a small number share the minor strain types. Subsequently, the unconfident predictions (i.e., grey-zone group) could result from the suboptimal learning. By contrast, there were only 10 MRSA isolates (17.9%) (Table 4) in the grey-zone group. According to other epidemiological investigations in Taiwan, the predominant clones of MRSA infection were ST59 and ST239, followed by ST45 and ST5 [26,27,28]. The emergence of MRSA ST8 (USA300) also gained much attention. With increasing prevalence since 2010, ST8 has become one of the major clones of MRSA infection in Taiwan [32]. The composition of predominant clones for MRSA is much simpler than that of MSSA. In this study, however, the MRSA isolates in the grey-zone group showed different sequence-type combinations. Uncommon types such as ST1232, ST6954, and ST1 account for 80% of MRSA in the grey-zone group (Figure 4). ST1232 is a single-locus variant of ST398 [33]. Both ST1232 and ST398 were clusters of CC398 MRSA. The ST1232 MRSA strain was related to Southeast Asia traveling, and ST398 was similar to European livestock-associated MRSA (LA-MRSA) [34]. Local transmission of the CC398 MRSA strain was still rare in Taiwan, and most of the cases were possibly livestock related [35]. According to a previous study, the predominant LA-MRSA clone in Taiwan is ST9 [36], also indicating the paucity of the CC398 strain in Taiwan. These uncommon sequence types of MRSA were less understood during model training, resulting in higher difficulty and uncertainty for the machine learning model in assigning preliminary AST.

## 5. Conclusions

Molecular characteristics are the key contributing factor in unfavored AST prediction by MALDI-TOF AI models. An uncommon sequence type of MRSA is more likely to have wrong results on preliminary AST. The genotypic diversity of MSSA is the main cause of the inferior prediction performance in the grey zone.

## Figures and Tables

**Figure 1 diagnostics-12-00413-f001:**
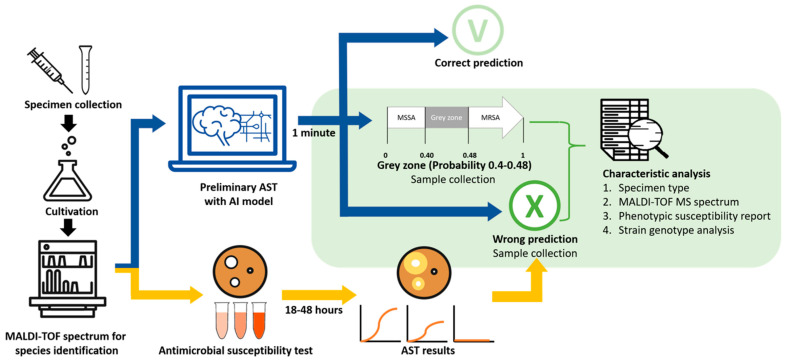
Schematic illustration of the study. The antibiotic susceptibility test (AST) is predicted by analyzing MALDI-TOF spectra with the artificial intelligence (AI) model. Based on the predictive probability generated by the AI model, cases with a probability of less than 0.4 were predicted as susceptible, whereas cases whose probability was larger than 0.48 were predicted as resistant. In contrast, the cases whose probabilities lie between 0.4 and 0.48 were defined as “grey zone”. In addition, cases whose predictive ASTs are different from final ASTs are categorized as “wrong prediction”, while those whose prediction matches final AST are “correct prediction”. We collected “grey zone” and “wrong prediction” cases for further analysis.

**Figure 2 diagnostics-12-00413-f002:**
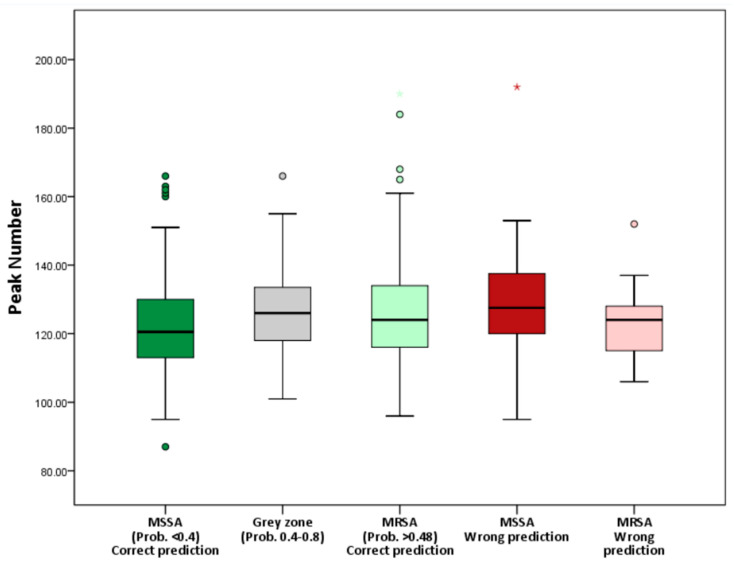
Number of peaks of the MALDI-TOF mass spectra in different groups. The number of peaks of different groups (i.e., MSSA with correct prediction, grey zone, MRSA with correct prediction, MSSA with wrong prediction, and MRSA with wrong prediction) shows no significant difference, indicating comparable quality of spectra between groups. Asterisk represents “extreme value”, which is more than 3 interquartile range (IQR) from the end of a box. Small circle represents “outliers”, which is more than 1.5 IQR but at most 3 IQR from the end of a box.

**Figure 3 diagnostics-12-00413-f003:**
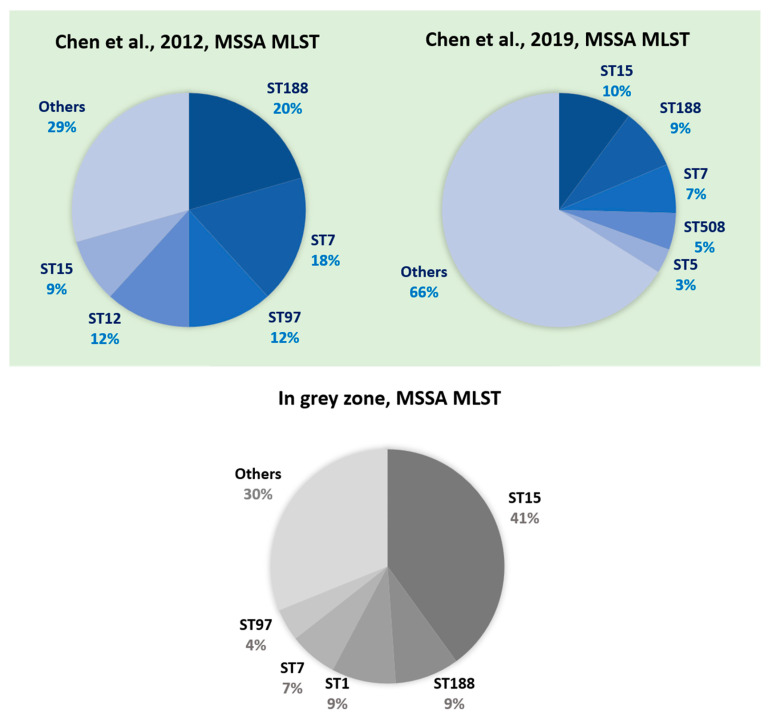
Distributions of multilocus sequence types for the MSSA isolates from Chen et al. [24], Chen et al. [25], and the grey-zone group in this study. The previous studies [24,25] showed that the common sequence types of MSSA infection in Taiwan were ST188, ST15, ST7, and ST97, but around 30–66% of MSSA isolates were other ST types. Similarly, in the grey-zone group of this study, MSSA strain types also show high diversity where 31% of MSSA isolates are not characterized in the top 5 common types (ST15, ST188, ST1, ST7, and ST97). Literature review of *S. aureus* epidemiology investigation in Taiwan is shown in Appendix A.

**Figure 4 diagnostics-12-00413-f004:**
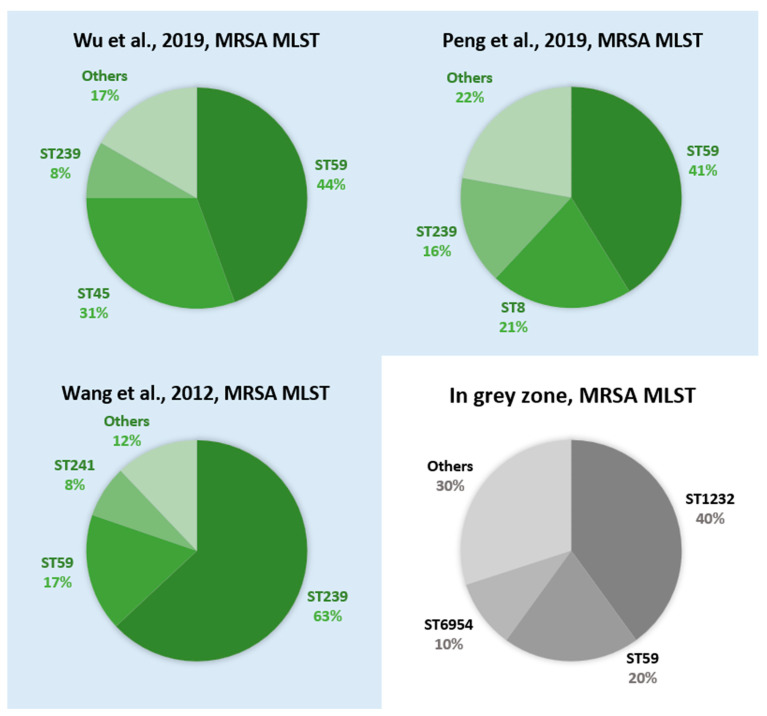
Multilocus sequence types of MRSA from Wu et al. [27], Peng et al. [28], Wang et al. [29], and the grey-zone group in this study. The previous studies [27,28,29] showed that the common sequence types of MRSA infection in Taiwan were ST59 and ST239, followed by ST45 and ST5. In this study, the MRSA isolates in the grey-zone group comprise uncommon sequence types such as ST1232, ST6954, and other types (accounting for 80%). Literature review of *S. aureus* epidemiology investigation in Taiwan is shown in Appendix A.

**Table 1 diagnostics-12-00413-t001:** Characteristics of the cases.

All *S. aureus*		Correct Prediction (*n* = 401)	Grey Zone (*n* = 56)	Wrong Prediction (*n* = 22)	*p*-Value
Specimen type (n (%))	Pus	216 (53.9%)	26 (46.4%)	15 (68.1%)	0.006 *
	Tissue	20 (5%)	2 (3.6%)	0 (0%)	
	Respiratory tract	76 (19.0%)	5 (8.9%)	1 (4.5%)	
	Blood	40 (10.0%)	11 (19.6%)	2 (9%)	
	Sterile body fluid	7 (1.7%)	5 (8.9%)	0 (0%)	
	Urinary tract	24 (6%)	2 (3.6%)	1 (4.5%)	
	Others	18 (4.5%)	5 (8.9%)	3 (13.6%)	
**MRSA (*n* (%))**		233 (58.1%)	10 (17.9%)	8 (30%)	
Specimen type (n (%))	Pus	117 (50.2%)	7 (70%)	6 (75%)	0.886
	Tissue	16 (6.9%)	0 (0%)	0 (0%)	
	Respiratory tract	46 (19.7%)	2 (20%)	0 (0%)	
	Blood	26 (11.2%)	1 (10%)	1 (12.5%)	
	Sterile body fluid	2 (0.9%)	0 (0%)	0 (0%)	
	Urinary tract	15 (6.4%)	0 (0%)	1 (12.5%)	
	Others	11 (4.7%)	0 (0%)	0 (0%)	
**MSSA (*n* (%))**		168 (41.9%)	46 (82.1%)	14 (70%)	
Specimen type (n (%))	Pus	99 (58.9%)	19 (41.3%)	9 (64.3%)	0.008 *
	Tissue	4 (2.4%)	2 (4.3%)	0 (0%)	
	Respiratory tract	30 (17.9%)	3 (6.5%)	1 (7.1%)	
	Blood	14 (8.3%)	10 (21.7%)	1 (7.1%)	
	Sterile body fluid	5 (3%)	5 (10.9%)	0 (0%)	
	Urinary tract	9 (5.4%)	2 (4.3%)	0 (0%)	
	Others	7 (4.2%)	5 (10.9%)	3 (21.4%)	

* *p*-Value < 0.05. The underline helps to identify which specimen type contribute most to the significant difference.

**Table 2 diagnostics-12-00413-t002:** Zone size of cefoxitin for MRSA and MSSA. MRSA with wrong AST prediction (predicted as MSSA) had a significantly larger zone diameter than correct-prediction and grey-zone groups, indicating that the phenotype of MRSA with wrong AST prediction is more like MSSA.

**MRSA**	**Correct Prediction (*n* = 190)**	**Grey Zone (*n* = 10)**	**Wrong Prediction (*n* = 6)**	** *p* ** **-Value**
Zone size (mm)	10.78 ± 3.49	11.4 ± 3.95	14.33 ± 3.01	0.004 *
**MSSA**	**Correct prediction (*n* = 150)**	**Grey zone (*n* = 34)**	**Wrong prediction (*n* = 11)**	** *p* ** **-Value**
Zone size (mm)	25.66 ± 2.04	25.53 ± 1.97	24.81 ± 1.72	0.113

* *p*-Value < 0.05.

**Table 3 diagnostics-12-00413-t003:** MIC of oxacillin using broth microdilution method. Both MSSA and MRSA had no significant difference in MIC between groups.

**MRSA**	**Correct Prediction (*n* = 31)**	**Grey Zone (*n* = 0)**	**Wrong Prediction (*n* = 1)**
Median MIC (µg/mL)	>4	NA	4
**MSSA**	**Correct prediction (*n* = 19)**	**Grey zone (*n* = 12)**	**Wrong prediction (*n* = 2)**
Median MIC (µg/mL)	0.5	0.5	0.375

**Table 4 diagnostics-12-00413-t004:** MLST of the *S. aureus* isolates in the grey zone.

	MRSA (*n* = 10)	MSSA (*n* = 46)
ST1232	4 (40%)	2 (4.3%)
ST59	2 (20%)	-
ST6954	1 (10%)	-
ST239	1 (10%)	1 (2.2%)
ST30	1 (10%)	1 (2.2%)
ST1	1 (10%)	4 (8.7%)
ST15	-	19 (41%)
ST188	-	4 (8.7%)
ST7	-	3 (6.5%)
ST97	-	2 (4.3%)
ST789	-	2 (4.3%)
ST6	-	1 (2.2%)
ST88	-	1 (2.2%)
ST182	-	1 (2.2%)
ST672	-	1 (2.2%)
ST2846	-	1 (2.2%)
ST2990	-	1 (2.2%)
None	-	2 (4.3%)

## Data Availability

The data presented in this study are available on request from the corresponding author. The data are not publicly available due to restriction policy of the institute.

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
