# Peer review of "Investigating Unfavorable Factors That Impede MALDI-TOF-Based AI in Predicting Antibiotic Resistance"

_diagnostics, 2022, doi:10.3390/diagnostics12020413_

Round 1

Reviewer 1 Report

  1. Some of the abbreviations are not defined the first time they appear
  2. in the result section the different statistical analyses would go better in subsections (3.1, 3.2...) instead of First, Second...
  3.  Table 1. As the age of the patients did not yield statistically significand differences, I think it could be left out from the table leaving only the specimens. 
  4. Table 1. For me it is not clear what the p-values represent. Please define more clearly. 
  5. Table 2. I believe that the zone size is in mm not cm. 
  6. Table 2 I do not agree with your conclusions. You cannot conclude that MRSA with 14.33 is closer to MSSA with 24-25 24-25 than the rest of MRSA with 10 or 11  
  7. Table 2  Why are less patients in the MSSA gray zone?   34 instead if 46. How were they selected?
  8. I do not understand table 3. It does not show on the schematic illustration of the study.
  9. The discussion section is confusing at some points. The conclusions do not derive from the results of the study. There are some valid observations, but needs more structuring. The study about enterococcus faecium I believe is not relevant since this study is about Staphylococcus. 

Reviewer 2 Report

  1. The novelty of this manuscript should be highlighted in the introduction section.
  2. Literature review is missing .some update in introduction and discussions should be discussed from 2019-2022
  3. Introduction part should be explore on targeted MALDI-TOF based AI in predicting antibiotic resistance
  1. The full form of all abbreviation should be mentioned initially…like MRSA??
  2. By what source was made in each plate colonies and what size in nanometer.
  3. What is the role of Desorption/Ionization onto the AI in predicting antibiotic resistance
  4. Any clinical relevance ?
  5. Discussion needs to improve, here you have to conclude major finding of your work instead of repeating introduction and abstract.

Round 2

Reviewer 2 Report

The reviewer has revised all comments. It has now been accepted for publication